# Plasticity and Potency of Mammary Stem Cell Subsets During Mammary Gland Development

**DOI:** 10.3390/ijms20092357

**Published:** 2019-05-13

**Authors:** Eunmi Lee, Raziye Piranlioglu, Max S. Wicha, Hasan Korkaya

**Affiliations:** 1Department of Biochemistry and Molecular Biology, Georgia Cancer Center, Augusta University, Augusta, GA 30912, USA; eunmil@princeton.edu (E.L.); RPIRANLIOGLU@augusta.edu (R.P.); 2Comprehensive Cancer Center, University of Michigan, Ann Arbor, MI 48109, USA; mwicha@med.umich.edu

**Keywords:** mammary stem cells, morphogenesis, epithelial plasticity, multipotent, unipotent

## Abstract

It is now widely believed that mammary epithelial cell plasticity, an important physiological process during the stages of mammary gland development, is exploited by the malignant cells for their successful disease progression. Normal mammary epithelial cells are heterogeneous and organized in hierarchical fashion, in which the mammary stem cells (MaSC) lie at the apex with regenerative capacity as well as plasticity. Despite the fact that the majority of studies supported the existence of multipotent MaSCs giving rise to both basal and luminal lineages, others proposed lineage restricted unipotent MaSCs. Consistent with the notion, the latest research has suggested that although normal MaSC subsets mainly stay in a quiescent state, they differ in their reconstituting ability, spatial localization, and molecular and epigenetic signatures in response to physiological stimuli within the respective microenvironment during the stages of mammary gland development. In this review, we will focus on current research on the biology of normal mammary stem cells with an emphasis on properties of cellular plasticity, self-renewal and quiescence, as well as the role of the microenvironment in regulating these processes. This will include a discussion of normal breast stem cell heterogeneity, stem cell markers, and lineage tracing studies.

## 1. Introduction

Normal tissues including the mammary gland consist of a cellular hierarchy; tissue specific (adult) stem cells (SC) at the apex with the ability to self-renew and generate all progeny, committed progenitors with a limited differentiation capacity, and terminally differentiated cells which constitute the functional gland. Adult SCs are distinguished from embryonic stem cells (ESC) in that their ability to generate progeny of distinct cell types is largely restricted to the particular organ from which SC originated. A series of studies, however, have challenged the notion of the traditional lineage restriction of organ specific SCs, demonstrating evidence that trans-differentiation of adult SCs into mature cells of different tissues is possible [1,2]. The discovery of induced pluripotent stem cells (iPSC) from a wide range of differentiated cell types [3,4] and subsequent studies demonstrating direct reprogramming of one adult cell into another functional cell [5,6,7,8,9] provided an indirect support for the possibility of trans-differentiation. Although the mechanism of action of trans-differentiation still remains elusive, it is clear that given the correct series of input signals, a cell can directly be pushed into a different cell type [5,6,7,8,9,10,11]. In normal development and under tissue homeostasis, lineage restriction of any adult stem cells is maintained by a lack of these unusual combinations of signals. However, we are beginning to appreciate that many of the experimental settings (transplant experiments) and pathological conditions (for example, tumor microenvironment) lead to cells experiencing a set of signals that induce a more stem-like state somewhere in between normal adult stem cells and iPSCs. Similarly the phenotypic and functional properties of mammary stem cells (MaSC) in mammary gland development are under intense investigation. Extensive studies thus far have proposed two opposing models of MaSCs. Majority of these studies have concluded that bipotent stem cells are able to give rise to both luminal and basal populations [12,13,14,15,16,17,18,19,20,21]. However, other studies proposed that the adult mammary gland only contains unipotent stem cells, each of which generate only luminal or basal lineages [20,21,22,23,24]. What these stem cells have in common though is that they have been shown to cross–communicate with their microenvironment to maintain homeostasis, which ensures the generation of mature functional cells throughout the life of organism without depletion of stem cell pools [12,25,26,27,28]. 

In tumors, the example of “vasculogenic mimicry”, by which malignant cells express endothelial specific genes in order to create cells and structures resembling to vasculature [29,30,31], reinforces the concept for an increased stem-like plasticity under pathological conditions. This is consistent with the notion that tumors may be not heterogeneous clonal populations, but are in effect “organ-like structures” composed of numerous cell types whose interaction promotes tumor progression and metastasis. 

In this review, we will focus on molecular mechanisms and clinical relevance of stem cell properties and their role during the organogenesis of mammary gland including the current state of stem cell markers. 

## 2. Lessons from Normal Mammary Gland Stem Cells 

The expansion and regeneration of mammary epithelium in reproductive cycles during puberty and pregnancy suggested that a self-renewing mammary stem cell population maintains the mammary gland. Early studies first describing the repopulation of mouse mammary gland via serial transplantation of retrovirally tagged epithelial fragments to demonstrate the clonal nature of the repopulation process [17,32,33] suggested that a subset of mammary epithelia may have self-renewing capacity. Initial experimental evidence was provided by demonstrating that a basal mammary epithelial cell with self-renewing capacity was able to generate a functional mammary gland in a murine model [16,18]. These studies were further corroborated by others using similar models [19,20]. Analogous to other well-studied organs such as the hematopoietic system [34] and gut [35], a similar hierarchical organization was found in the mammary gland [18,36,37,38,39]. However, MaSCs were enriched from the basal compartment of the mammary gland but carried the impurities of progenitors and myoepithelial cells which thus contributed to a conflicting interpretation of data when used in functional characterization studies. The majority of these conclusions were extrapolated from studies using the murine mammary gland due to accessibility and ease of generating transgenic models. Although we now know that there are important structural, hormonal and developmental discrepancies between human and mouse mammary glands [26,40], overwhelming evidence supported the physiological and functional similarities of multipotent MaSCs in both systems. Therefore, a significant progress was made in understanding of the mammary gland development and malignant transformation of mammary epithelia using the murine mammary gland as well as the transgenic models.

## 3. Long-Lived Quiescent MaSCs Which Are Activated in Response to Stimuli

Although both lineage tracing and transplantation assays supported the existence of bipotent embryonic MaSCs, whether these bipotent embryonic stem cells are maintained in adult mammary gland remains controversial [19,20,41]. Recent studies have explored the idea of quiescent stem cells as discovered in other systems such as hematopoietic system [42], and reported the existence of long-lived, quiescent MaSCs which re-enter the cell cycle and give rise to both lineages in response to stimuli in adult mammary gland [12,13,41,43]. A fluorescent dye, PKH-26 which binds to cell membranes and segregates in daughter cells in each cell division, was used to determine the rate of cell division in mammosphere forming basal cells and repopulating efficiencies in transplantation assays [43]. Interestingly, only slow-cycling PKH^hi^ cells (0.5–1%), sorted from secondary mammospheres, were able to reconstitute mammary gland, while no mammary reconstitution was observed in mice transplanted with PKH^+^ or PKH^lo^ cells isolated similarly from secondary mammospheres [43]. Two subsequent studies corroborated these findings, supporting the evidence of quiescent MaSCs becoming activated in adult mammary glands upon stimuli. In an effort to characterize basal compartment, Fu et al. performed gene expression profiling of Lgr5+ cells and identified Tetraspanin8 (*Tspan8*) as a surface marker [12]. Fractionation based on the Lgr5 and Tspan8 expressions (Lgr5^+^Tspan8^hi^, Lgr5^−^Tspan8^hi^, Lgr5^+^Tspan8^−^ and Lgr5^−^Tspan8^−^) revealed that although all subsets displayed varying degrees of clonogenic potential in vitro and repopulating capacity in transplantation assays, ductal outgrowths were not identical. Lgr5^+^Tspan8^hi^ cell population not only exhibited a superior repopulating capacity, but also gave rise to all lineages. Interestingly, Lgr5^+^Tspan8^hi^ subset consisted of slow cycling cells with a distinct epigenetic profile [12]. Furthermore, these otherwise quiescent MaSCs were shown to be activated by steroid hormones as evidenced by reduced proportion of quiescent (Lgr5^+^Tspan8^hi^) subset in mammary glands of pregnant mice compared to those of virgin mice [12]. Together the data reveal the existence of quiescent MaSC subsets which may be activated upon stimuli in adult mammary glands (Figure 1). 

Employing single cell gene expression analyses, Cai et al. identified a quiescent cell population which expressed high levels of *Bcl11b* and was restricted to the basal compartment [13]. The clonogenicity and transplantation assays of *Bcl11b*^hi^ and *Bcl11b*^low^ subsets within the basal Lin^-^CD49f^hi^CD24^med^ compartment, demonstrated that *Bcl11b*^hi^ subset was substantially more efficient in generating colonies and engraftment capacity. Furthermore, conditional deletion of *Bcl11b* in mammary gland resulted in significant reductions of postnatal mammary gland development due to exhaustion of ductal mammary epithelium [13]. When compared with *Procr*^hi^ cell subset [14], *Bcl11b* marked a distinct cell population within the basal compartment with CD49f^hi^CD24^med^ phenotype [13]. Whereas *Procr*+ cells were actively cycling, *Bcl11b*^+^ cells were mainly quiescent and that this gene functionally regulated the homeostasis of quiescent basal cell population [13]. Although quiescent and cycling MaSCs in human mammary glands has not been well defined, so far the studies have established the EpCAM^−^CD49f^hi^ phenotype being the marker of human MaSCs [44,45,46]. In addition human multipotent and basal progenitors expressed surface markers such as CD10, CD90 and TP63 [44,46]. Besides these markers, the expression of enzymatic activity of ALDH1 has also been shown to label both human and mouse multipotent MaSC and luminal progentors [39,47,48,49]. Furthermore, studies established that both mouse and human MaSCs and basal progenitors lack ER expression (ER-) while luminal progenitors show low levels of ER (ER+/−) expression (Figure 1) [45,46,50]. However, in mouse mammary gland CD49f^hi^Sca-1^+^ population was primarily ER-positive (ER+) as previously shown [51]. Together these studies provide a strong evidence for hierarchical organization of the mammary gland and the quiescent MaSCs at the apex which are important for repopulating and long-term maintenance of the mammary gland.

## 4. Competing Models: Multipotency and Lineage-Restricted Unipotent MaSCs

Technological advances and extended knowledge of stem cell markers have allowed better interrogation of single MaSCs in spatiotemporal studies. With these advanced studies, two competing concepts emerged on the functional characterization of MaSCs, namely whether the mammary gland consisted of bipotent or unipotent stem cells (Figure 2). The widely accepted bipotent MaSC model proposes that there exists a single stem cell population that can give rise to all lineages of the mammary tissue and coordinate ductal epithelial expansion as well as the maintenance of the adult mammary gland. Whereas the lineage-restricted unipotent MaSC model postulates that each compartment is maintained by its own unipotent stem cells in the adult mammary gland. Although experimental designs and approaches may partially account for the fundamental discrepancies of the two models, alternative explanations such as the existence of long-lived progenitors [15] or more than one type of bipotent MaSCs [12,13] could also help to explain the degree of complexity seen within the adult mammary gland. 

In support of the bipotent MaSCs in morphogenesis and adult gland maintenance, basal Lin^-^CD29^hi^CD24^+^ and luminal Lin^−^CD29^lo^CD24^+^ cell compartments, segregated by fluorescence-activated cell sorting (FACS) were tested in transplantation assays. The results showed that the basal cells could repopulate functional mammary glands, however, luminal cells lacked this capacity [16,17]. These studies provided early evidence that the basal compartment with a Lin^−^CD29^hi^CD24^+^ phenotype is enriched for MaSCs. 

Utilizing a clonal cell-fate mapping study, it was reported that both MaSCs and long-lived progenitor cells drive the morphogenesis during puberty, however, only MaSCs contributed to remodeling of the adult mouse gland and homeostasis of mammary ducts [15]. These experiments involved the use of a stochastic multicolor *cre* reporter combined with three-dimensional imaging in order to demonstrate that Elf5-expressing luminal progenitors contributed to the morphogenesis during puberty. Subsequent studies identified a subset of cells within the basal Lin^−^CD29^hi^CD24^+^ compartment which were characterized by protein C receptor (*Procr*) expression [14]. While *Procr* expression labels basal and stromal cells, its expression lacks in the luminal compartment. In transplantation studies, Procr^+^ CD29^hi^CD24^+^ cells compared to Procr^-^CD29^hi^CD24^+^ cells, showed a fivefold higher repopulating efficiency. Interestingly, RNA-seq analyses revealed that Procr^+^ cells exhibit a higher expression of epithelial-mesenchymal transition (EMT) related genes such as N-cadherin, Zeb1, and Zeb2, and lower expression of epithelial signature genes such as EpCAM, E-cadherin, and claudins [14]. This finding hints the possibility of a rare population of highly plastic MaSCs and has important implications for cancer initiation [52]. 

However, other studies have indicated that bipotent stem cells were only found during early development, and that under post-natal development and adult homeostasis, luminal and basal compartments each have their own separate stem cells. In support of this hypothesis, Van Keymeulen et al. performed advanced lineage-tracing experiments in embryonic, adult, pregnant, and involuting mammary glands. These experiments utilized K14rtTA/TetO-Cre/Rosa-YFP and K5-CreER/Rosa-YFP mice to track basal stem cells while K8-CreER/Rosa-YFP and K18-CreER/Rosa-YFP mice were used to track luminal stem cells [22]. Induction of K14-positive YFP cells resulted in labelling of both basal and luminal cells during morphogenesis suggesting that embryonic K14 cells are multipotent. However, postnatal induction of K14+ or K5+ YFP cells displayed exclusive labelling of basal cells that clonally expanded during puberty and pregnancy. In contrast, clonal analyses of K8+ YFP cells in postnatal mice revealed an expansion of the luminal lineage in adult life and pregnancy, while induction of K18+ YFP cells resulted in the less efficient generation of the luminal lineage with a reduced potency and thus K18-labelled cells were considered as the more committed progenitors. However, further studies based on lineage tracing at saturation using doxycycline showed that only unipotent mammary stem cells generate mammary gland and tissue remodeling [53].

Leucine-rich repeat-containing G-protein coupled receptor 5 (Lgr5), an intestinal stem cell marker [54], was shown to be expressed predominantly within the basal compartment and a small fraction within the luminal cell population [22]. Although Lgr5 expression was independently shown to mark embryonic MaSCs, it was dispensable for both mammary and adult MaSC activity as well as tumorigenesis [55]. 

Undoubtedly, “lineage-tracing” experiments have proven to be valuable in addressing the origin of the stem cell and tracing subsequent lineages in morphogenesis and maintenance of the adult mammary gland. However, previous studies primarily utilizing serial transplantation assays had shown a significant contribution of multipotent progenitors that were able to reconstitute both luminal and myoepithelial lineages [16,18,22]. In order to rectify the differences in these experiments, Van Keymeulen et al. transplanted mixtures of YFP-labelled basal cells together with unlabeled luminal cells and showed the regenerated mammary gland in which the vast majority of YFP+ cells were basal, while only rare clones expressed YFP in the luminal compartment [22]. Similarly, transplantation of YFP-labelled luminal cells with unlabeled basal cells resulted in the regeneration of new mammary glands consisting of YFP-labelled cells exclusively within the luminal compartment. These results helped to resolve the discrepancies in experiments by showing that while regeneration of a mammary gland in the transplantation assay is predominantly driven by lineage restricted progenitors, the basal compartment does have increased plasticity while the luminal cells remained restricted. Combined, these experiments lead to a model where the early development of the mammary gland is driven by multipotent progenitors, but during homeostasis, lineage restricted unipotent stem cells likely maintain the separate lineages. However, disorganization of the mammary tissue through injury, transplantation assays, or even cancer can lead to increased plasticity of the restricted progenitors back to a multipotent state.

## 5. Mammary Stem Cell Plasticity Regulated by the Microenvironment during Mammary Development

Epithelia in general show enormous heterogeneity and functional plasticity throughout the body [25]. Although mammary epithelium is the functional unit of the mammary gland, its development and maintenance also requires a complex cross-talk with the surrounding stroma [26]. Understanding this cross-talk has been proven to be a challenging task due to complexities of the microenvironment in which mammary epithelia go through morphogenesis, lactation and involution in each cycle of pregnancy. One such example of stroma contributing the SC plasticity is demonstrated in hair regeneration. In resting hair follicles, the bulge is comprised of heterogeneous stem cell population, while CD34 expression marks quiescent SCs in both upper and lower bulge, Lgr5 expressing and actively cycling SCs are restricted to the lower bulge [56]. A recent study reported that the elimination of the cycling Lgr5+ cells by using diphtheria-toxin-mediated cell ablation abrogated the hair regeneration but it was reversed during the recovery phase by activation of quiescent CD34^+^ stem cells via the inflammatory responses [57]. The fact that the activation of otherwise quiescent CD34^+^ stem cells by inflammatory responses provides a compelling evidence for the role of the microenvironment in regulating stem cell plasticity and tissue homeostasis. In line with these studies, it has been suggested that inflammatory responses and microenvironmental cues may lead to stem cell plasticity leading to heterogeneous MaSC phenotypes [52]. Interestingly, when FACS-sorted YFP+ basal and luminal cells were co-transplanted, only basal cells gave rise to both basal and luminal lineages. In order to explain the discrepancy of the data, Ven Keymeulen and colleagues argued that the experimental setting of co-transplantation forces this multi-lineage differentiation of basal cells [22]. In line with the notion, although it is yet to be experimentally determined, it may entirely be possible that transplantation assay may potentially induce inflammatory responses which in turn activate otherwise quiescent MaSCs [12,13] thereby regenerating the mammary gland. In addition, emerging studies implicate a secondary advantage of the MaSC plasticity or remaining quiescent that provides protection from pathological and immunological insults [58,59]. Type I interferons (IFN-1) induces a transient HSC proliferation in acute inflammation, however, in response to chronic IFN-1 exposure, HSCs rapidly return to quiescence [59]. This reestablished quiescence protects HSCs from IFN-1-induced exhaustion unless forced back into the cell cycle due to in vitro culture or transplantation conditions [59]. A similar mechanism was reported to protect normal and malignant mammary stem cells from the cytotoxic effect of IFN via miR-199a-mediated repression of nuclear receptor corepressor LCOR [58]. It was demonstrated that elevated miR-199 expression and subsequent LCOR repression protects stem cells from differentiation and senescence induced by IFNs that are produced by epithelial and immune cells. Consistent with the notion, quiescent MaSCs located at the proximal region of ductal tree were protected in Lgr5-GFP-IREScreER^T2^/R26R-tdTomato mice during involution [12]. 

Although there is yet to be a consensus on the location of quiescent MaSCs in the mammary ductal tree [12,13,60], a distinct location and surrounding microenvironment may also play a role in maintaining them in quiescence [12] (Figure 2). 

## 6. Markers of Mammary Stem Cells and Progenitors

It is well-established that embryonic MaSCs give rise to mammary epithelium during postnatal development. Early transplantation studies utilized tissue fragments of the mammary gland to demonstrate the capacity of certain fractions to form an entire gland when transplanted into an epithelium-free mammary fat pads [61]. Single cell suspension and limiting-dilution transplantation assays further corroborated these findings and demonstrated the capacity of a single cell to regenerate entire functional mammary gland [17]. Subsequently, FACS-based segregation of cells enabled the investigators to interrogate single cells. Because mammary gland comprises of different cell types, stroma (non-epithelial cells) were eliminated by using antibodies against endothelial (CD31) and hematopoietic (CD45, TER119) antigens called in general Lin^-^ [16]. There were four distinct Lin^-^ cell populations based on the expressions of surface antigens, CD29 (β1-integrin), identified in skin stem cells [62] and CD24, a glycoprotein expressed in many tumors [63] and used to enrich neural stem cells [64]. When frequency of mammary repopulating unit (MRU) was tested, Lin-CD29^hi^CD24^+^ subset displayed an eightfold enrichment over the Lin^-^ population, whereas the other three subsets showed no significant enrichment [16]. In the human mammary gland, this basal compartment with MRU capacity has been characterized with the EpCAM^low^-CD49f^hi^ phenotype [45,46]. Similarly, further studies determined that CD49f may also be a marker of mouse MaSCs. Furthermore, Lin-CD29^hi^CD24^+^ subpopulation gave rise to quiescent and asymmetrically dividing cells as evidenced by the presence of long-term label-retaining cells. Stingl et al. reported an MRU enrichment within Lin^-^ subset expressing high CD49f (α6-integrin) and low Sca-1 (CD49f^hi^Sca-1^low^) which also constituted the ER+ subset of the luminal progenitors [18,51]. Interestingly, CD49f expression was correlated with CD29^hi^CD24^+^ phenotype [16]. Moreover, when human mammary epithelial cells fractionated based on the CD49f and EpCAM (CD326) and MRU activity was tested, CD49f^+^EpCAM^neg-low^ subpopulation showed significantly higher enrichment compared to CD49f^-^EpCAM^-^ subset [65]

Although Sca-1^+^ cell population was reported to have enriched MRU activity [66], it was not independently corroborated by other investigators [16,18]

A “side population” of cell subset segregated by the efflux of the Hoechst 33342 and Rhodamine (Rho) dyes was demonstrated to have stem cell activities from various adult tissues including breast [67]. However, subsequent studies [16,18,68] were not able to validate the MRU activity of “side population” and that may partially be explained by Hoechst 33342-related toxicities [69]. 

In genetic “lineage tracing” studies, embryonic Keratin 14 and 5 (K5/14) expressing cells were shown to have bipotent MaSC activity while K8/18 expressing cells to have unipotent luminal SC activity in adult mammary glands [22]. 

Activity of aldehyde dehydrogenase 1 (ALDH1), a detoxifying enzyme, was shown to mark HSCs and play a role in their engraftment capacity [70,71]. Consistent with these findings, the ALDH1 enzymatic activity as measured by Aldefluor assay enriches normal human mammary stem cells as evidenced by their ability to generate human mammary ductal structures comprised of the luminal and basal layers in humanized mouse model [39]. 

## 7. Concluding Remarks

As an emerging research becomes more focused on the properties of mammary stem cells such as epithelial plasticity, quiescence, and self-renewal, the importance of these for the tumor development and progression becomes ever more critical. In our review, we covered the current data on the complexity and importance of epithelial cell plasticity, quiescence, and cellular heterogeneity in mammary gland development. It has become more evident that these properties are exploited by the malignant cells during tumor progression and metastasis. Therefore, information learned from MaSCs and normal mammary gland development may provide unique opportunities for future therapy to target plasticity of stem-like tumor cells.

## Figures and Tables

**Figure 1 ijms-20-02357-f001:**
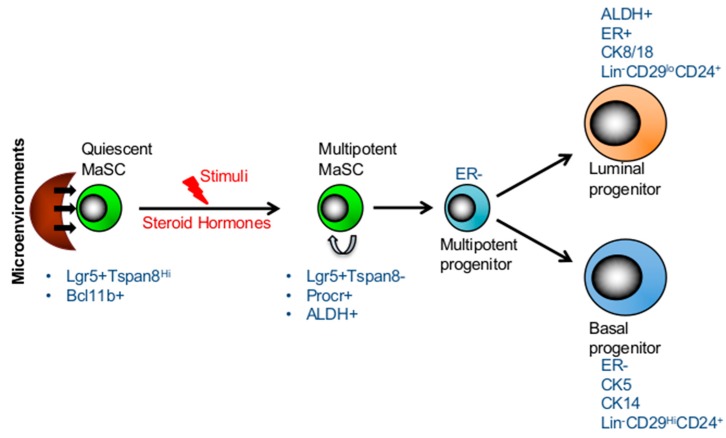
Hierarchical structure and plasticity of mammary stem cells. The transition between long-lived, quiescent, and proliferating MaSCs is regulated by its microenvironment. Quiescent MaSCs can re-ender the cell cycle and give rise to both lineages in response to stimuli, such as steroid hormones in the adult mammary gland. The markers of two subsets of stem cells and progenitors are presented in the figure.

**Figure 2 ijms-20-02357-f002:**
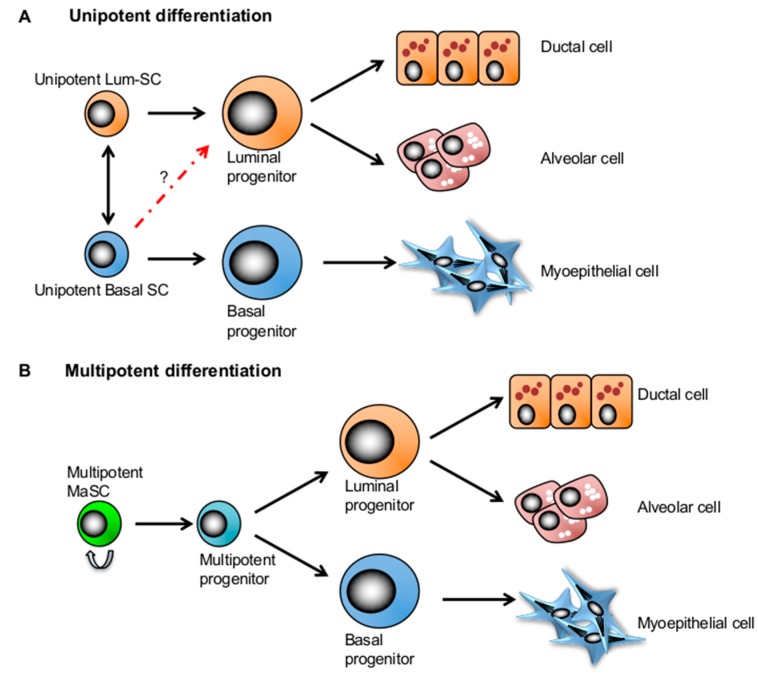
Two competing mammary stem cell (MaSC) models in mammary gland development. (**A**) The lineage-restricted unipotent MaSC model postulates that each compartment is maintained by its own unipotent stem cells in the adult mammary gland. (**B**) The bipotent MaSC model proposes that there exist a multipotent stem cell population that can give rise to all lineages of the mammary tissue.

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
