# Peer review of "Plasticity and Potency of Mammary Stem Cell Subsets During Mammary Gland Development"

_ijms, 2019, doi:10.3390/ijms20092357_

Round 1

Reviewer 1 Report

Lee et al. explains that lineage tracing has become a reliable approach to study cellular hierarchies. Lineage tracing studies are established by using mainly cytokeratin promoters and other promoters and transplanted mammary tissues has been used to identify the capability of each subtype of mammary gland cell to reconstitute a new full gland. Studies on mammary gland development reveal that differentiation of multipotent stem cell starts during embryogenesis and unipotent stem cells in adulthood. Some studies have focused on studying the implication of the plasticity of mammary cells mainly due to inflammatory stimuli.

This review is complete and the figures are well designed, however I have some minor comments below.

In general, the review paper is well constructed and acknowledgeable. Each section has distinct purpose and the topic is interesting.

Other comments:

1.- The authors should include the reference of Wuidart et al. (Genes Dev. 2016). In that study the authors showed how saturating doses of doxycycline, which allow labeling 100% of the luminal or basal compartment, gave rise to a progeny that remains unipotent reinforcing the model that the mammary homeostasis is maintained by unipotent stem cells in adulthood.

2.- The authors could also include the reference Rodilla et al. (Plos Biol 2015). In that study the authors showed how the expression of CD49f/Sca1+ correlates to the ER+ subpopulation within the luminal compartment with reduced clonogenic capacity, similarly to Stingl et al. It would be interesting to mention it specifically for the Markers of mammary SC and progenitors section in order to define markers for the subpopulations in the luminal compartment.

3.- In the Figure1, if I understood correctly, they want to show the two current models regarding mammary gland hierarchy. It is unclear (to me) what does it mean the arrow between both luminal and basal unipotent cells. I guess it is to represent the plasticity? They do not comment what the red arrow means in the unipotent model. In my opinion, they should remove the red arrow, since the second model explores the other possibility, that BCs are multipotent giving rise to both luminal and basal progenitors.

Author Response

Q:The authors should include the reference of Wuidart et al. (Genes Dev. 2016). In that study the authors showed how saturating doses of doxycycline, which allow labeling 100% of the luminal or basal compartment, gave rise to a progeny that remains unipotent reinforcing the model that the mammary homeostasis is maintained by unipotent stem cells in adulthood.

A: We are grateful for the constructive comments. We now included this data and cited the article in our revised manuscript. 

Q: The authors could also include the reference Rodilla et al. (Plos Biol 2015). In that study the authors showed how the expression of CD49f/Sca1+ correlates to the ER+ subpopulation within the luminal compartment with reduced clonogenic capacity, similarly to Stingl et al. It would be interesting to mention it specifically for the Markers of mammary SC and progenitors section in order to define markers for the subpopulations in the luminal compartment.

A: We thank the reviewer for pointing out this important information, we now included this data and cited the paper in our revised manuscript. 

Q: In the Figure1, if I understood correctly, they want to show the two current models regarding mammary gland hierarchy. It is unclear (to me) what does it mean the arrow between both luminal and basal unipotent cells. I guess it is to represent the plasticity? They do not comment what the red arrow means in the unipotent model. In my opinion, they should remove the red arrow, since the second model explores the other possibility, that BCs are multipotent giving rise to both luminal and basal progenitors.

A: We agree with the reviewer that the red line was confusing, as suggested we now removed the red arrow and updated our figure 1. 

Reviewer 2 Report

The review is well written and summarize the most important articles published in the field of mammary gland biology;

Only some observations:

Personally, I do not think that Luminar progenitor cells (CD49+/EpCAM+) in the mammary gland can be considered totally Estrogen Receptor Positive as indicated in the Figure 2, they express some, but very few compared to luminal cells (Lim et al., 2009, , Tornillo G. et al., 2018,. Cell Reports). I would indicate them as ER-/+;

Figure 2 may be completed with Luminal and myoepithelial differentiated cells with markers; I suggest to the authors to prepare two different graphs, one for mouse-based lineage and human-based lineage. Most of the markers shown are mouse ones, but other mentioned in the text as CD49f+EpCAM-/low (Lim et al., 2011) are not indicated in the figure, (which is human marker). 

ALDH+ cells  have been shown to have mammary stem cell properties when transplanted in mouse (Ginestier et al., 2007) but ALDH+ cells are also considered as a luminal progenitor cell marker in the human mammary gland (Eirew et al., 2012; Sheata et al., 2012; Domenici et al., 2019). 

The paragraph "Markers of mammary stem cells and progenitors", talked about markers which identify mammary stem cells and progenitors as: CD49f+EpCAM-/low and ALDEFLUOR positive cells. I would update that paragraph with recent bibliography,about markers/pathways that characterize/maintain that population, in which a lot of recent literature has came out. 

Author Response

Q: Personally, I do not think that Luminar progenitor cells (CD49+/EpCAM+) in the mammary gland can be considered totally Estrogen Receptor Positive as indicated in the Figure 2, they express some, but very few compared to luminal cells (Lim et al., 2009, , Tornillo G. et al., 2018,. Cell Reports). I would indicate them as ER-/+;

A: We grately appreciate reviewers constructive comments. We agree with the reviewer that a subset of the luminal progenitors express ER not the entire population as previously indicated that the ER+ subset was contained within the CD49fhiSca-1+ population as previously reported by Rodilla et al 2015, we now included these details in our revised manuscript. we also included those indicated literatures in our revised manuscript. 

Q: Figure 2 may be completed with Luminal and myoepithelial differentiated cells with markers; I suggest to the authors to prepare two different graphs, one for mouse-based lineage and human-based lineage. Most of the markers shown are mouse ones, but other mentioned in the text as CD49f+EpCAM-/low (Lim et al., 2011) are not indicated in the figure, (which is human marker). 

A: We thank reviewers comments, we now included human markers in figure 2, however, we think that addition of differentiated luminal and myoepithelial cells in this figure is beyond the scope of the manuscript. 

Q: ALDH+ cells  have been shown to have mammary stem cell properties when transplanted in mouse (Ginestier et al., 2007) but ALDH+ cells are also considered as a luminal progenitor cell marker in the human mammary gland (Eirew et al., 2012; Sheata et al., 2012; Domenici et al., 2019). 

A: We now included these information and cited the articles in revised manuscript. 

Q: The paragraph "Markers of mammary stem cells and progenitors", talked about markers which identify mammary stem cells and progenitors as: CD49f+EpCAM-/low and ALDEFLUOR positive cells. I would update that paragraph with recent bibliography,about markers/pathways that characterize/maintain that population, in which a lot of recent literature has came out. 

A: we appreciate the comment by the reviewer, however, we aimed to focus on the main theme of the manuscript and addition of those pathways will dilute the main focus of the manuscript.